# Pan-cancer landscape of homologous recombination deficiency

Luan Nguyen [1], John W. M. Martens [2,3], Arne Van Hoeck [1,5] & Edwin Cuppen [1,4,5✉]

Homologous recombination deficiency (HRD) results in impaired double strand break repair and is a frequent driver of tumorigenesis. Here, we develop a genome-wide mutational scar-based pan-cancer Classifier of HOmologous Recombination Deficiency (CHORD) that can discriminate *BRCA1*- and *BRCA2*-subtypes. Analysis of a metastatic ($n = 3,504$) and primary ($n = 1,854$) pan-cancer cohort reveals that HRD is most frequent in ovarian and breast cancer, followed by pancreatic and prostate cancer. We identify biallelic inactivation of *BRCA1*, *BRCA2*, *RAD51C* or *PALB2* as the most common genetic cause of HRD, with *RAD51C* and *PALB2* inactivation resulting in *BRCA2*-type HRD. We find that while the specific genetic cause of HRD is cancer type specific, biallelic inactivation is predominantly associated with loss-of-heterozygosity (LOH), with increased contribution of deep deletions in prostate cancer. Our results demonstrate the value of pan-cancer genomics-based HRD testing and its potential diagnostic value for patient stratification towards treatment with e.g. poly ADP-ribose polymerase inhibitors (PARPi).

[1] Center for Molecular Medicine and Oncode Institute, University Medical Center Utrecht, Utrecht, The Netherlands. [2] Department of Medical Oncology, Erasmus MC Cancer institute, Erasmus University Medical Center, Rotterdam, The Netherlands. [3] Center for Personalized Cancer Treatment, Rotterdam, The Netherlands. [4] Hartwig Medical Foundation, Amsterdam, The Netherlands. [5]These authors jointly supervised this work: Arne Van Hoeck, Edwin Cuppen. ✉email: ecuppen@umcutrecht.nl

The homologous recombination (HR) pathway is essential for high-fidelity DNA double strand break (DSB) repair and involves numerous genes including *BRCA1* and *BRCA2*. HR deficiency (HRD) due to inactivation of such genes leads to increased levels of genomic alterations[1]. HRD is a common characteristic of many tumors and is frequently observed in breast and ovarian cancer[2]. Accurate detection of HRD is of clinical relevance as it is indicative of sensitivity to targeted therapy with poly ADP-ribose polymerase inhibitors (PARPi)[3,4] as well as to DNA damaging reagents[1].

In the clinic, germline *BRCA1/2* mutation status is currently the main genetic biomarker of HRD[5]. However, germline testing has its drawbacks: (i) it is dependent on the completeness and accuracy of clinical variant annotation databases (e.g. ClinVar); (ii) epigenetic silencing is overlooked; (iii) partial/complete deletions of the *BRCA1/2* loci are missed by current clinical genetic testing, resulting in *BRCA1/2* status reporting based on the wild type allele from contaminating normal tissue; and (iv) HRD can be driven purely by somatic events. Furthermore, the focus on *BRCA1/2* overlooks inactivation of other HR pathway genes. Consequently, patients may receive incorrect treatment or miss out on treatment opportunities, thus necessitating the development of better biomarkers for HRD.

It was recently shown that somatic passenger mutations, which are identified efficiently by whole-genome sequencing (WGS), can provide insights into the mutational processes that occurred before and during tumorigenesis, paving the way for novel opportunities for clinical tumor diagnostics[6]. For the repair of DSBs, HRD tumors are dependent on alternative more error-prone pathways including microhomology mediated end-joining (MMEJ)[7], resulting in a characteristic mutational footprint across the genome that can be used to detect HRD regardless of the underlying cause (whether genetic or epigenetic). Indeed, some mutational footprints were found to be associated with *BRCA1/2* deficiency, namely deletions with flanking microhomology, as well as several "mutational signatures" including two COSMIC single nucleotide variant (SNV) signatures and two structural variant (SV) signatures[8]. These features were used to develop a breast cancer-specific predictor of HRD known as HRDetect[9]. Application of this tool in primary tumors revealed that the prevalence of HRD extends beyond *BRCA1/2*-deficient breast cancer tumors, and occurs at varying frequencies in different cancer types[10]. However, HRD rates in advanced metastatic cancer remain unclear, although these are the patients that are increasingly targeted with personalized treatments including PARP inhibitors for *BRCA*-deficiency[5].

Here, we describe the development of a random forest-based Classifier of HOmologous Recombination Deficiency (CHORD) for pan-cancer HRD detection. With this model, we demonstrate that accurate prediction of HRD is possible across cancer types using specific SNV, indel, and SV types. We identify inactivation of *BRCA1*, *BRCA2*, *RAD51C*, and *PALB2* as the most frequent genetic cause of HRD pan-cancer in both primary and metastatic cancer, with the latter two genes resulting in the same mutational footprints as *BRCA2* (consistent with the findings of recent studies in breast cancer[11,12]). In addition, we find that the underlying genetic inactivation of these genes is cancer type specific, but independent of tumor progression state.

## Results

### Random forest classifier training

For the development of CHORD, we used WGS data of 3824 solid tumors from 3584 patients from the pan-cancer metastatic cohort of the Hartwig Medical Foundation (HMF)[13]. From these, we selected tumor samples with biallelic loss of *BRCA1* or *BRCA2*, and non-mutated *BRCA1/2*, to obtain a high confidence set of samples belonging to three classes for classifier training (*BRCA1*-deficient, *BRCA2*-deficient, and *BRCA1/2*-proficient). To this end, we screened each sample to identify those samples with one of the following events in *BRCA1/2*: (i) complete copy number loss (i.e. deep deletion), (ii) loss-of-heterozygosity (LOH) in combination with a pathogenic germline or somatic SNV/indel (as annotated in ClinVar, or a frameshift), or (iii) 2 pathogenic SNV/indels. This unbiased approach revealed 35 and 89 samples with *BRCA1* or *BRCA2* biallelic loss of function, respectively, which were labeled as HRD for the training. Conversely, 1,902 samples were labeled as HR proficient (HRP) as these samples were observed to carry at least one functional allele of *BRCA1/2*. In total, 2026 out of 3824 samples (53% of the HMF dataset) were used to train the classifier (Supplementary Fig. 1).

The occurrence of three main somatic mutation categories were used as features for training (Fig. 1a), which included (i) SNVs subdivided by base substitution type; (ii) indels stratified by the presence of sequence homology, tandem repeats, or the absence of either; and (iii) structural variants (SV), stratified by type and length. An initial feature analysis revealed that small deletions with ≥2 bp flanking homology were together more predictive of *BRCA1/2* deficiency versus deletions with 1 bp flanking homology (Supplementary Fig. 3). Thus, deletions with flanking homology were further split into these two homology length bins. The occurrence of the 29 features together formed a contribution profile for each sample. From this, relative contributions per mutation category were calculated to account for differences in mutational load across samples (Fig. 1a). These features are henceforth collectively referred to as "mutation contexts".

A random forest was then trained to predict the probability of *BRCA1* or *BRCA2* deficiency (Fig. 1b). Briefly, a core training procedure performed feature selection and class resampling (to alleviate the imbalance between the three classes). This core procedure was subjected to 10-fold cross-validation (CV) which was repeated 100 times to filter samples from the training set that were not consistently HRD or HRP. A sample was considered HRD if the sum of the *BRCA1* and *BRCA2* deficiency probabilities (henceforth referred to as the HRD probability) was ≥0.5. This core procedure was reapplied to the filtered training set to yield the final random forest model which we refer to as "CHORD" (Supplementary Figs. 2a, b and 4).

The presence of deletions with ≥2 bp flanking homology (del. mh.bimh.2.5) was found to be the most important predictor of HRD. Additionally, CHORD uses 1–10 kb and to a lesser extent 10–100 kb duplications (DUP_1e03_1e04_bp and DUP_1e04_1e05_bp, respectively) for distinguishing *BRCA1* from *BRCA2* deficiency. Given that deficiencies in other HR genes may lead to similar phenotypes, we have coined the terms "*BRCA1*-type HRD" and "*BRCA2*-type HRD" to describe these HRD subtypes (Fig. 1c). Together, the features that are predictive of HRD are in line with those of a previously described HRD classifier HRDetect[9]. However, the feature weights differ markedly likely due to differences in the background mutational landscape between the pan-cancer cohort used here compared to the breast cancer cohort used for training HRDetect.

### Performance of CHORD

Two approaches were used to assess the performance of CHORD. In the first approach, 10-fold CV was performed on the training data which allows every sample to be excluded from the training set, after which unbiased HRD probabilities can be determined (Supplementary Fig. 2c). The probabilities of all prediction classes (i.e. HRD, *BRCA1*-type HRD, *BRCA2*-type HRD) were highly concordant with the

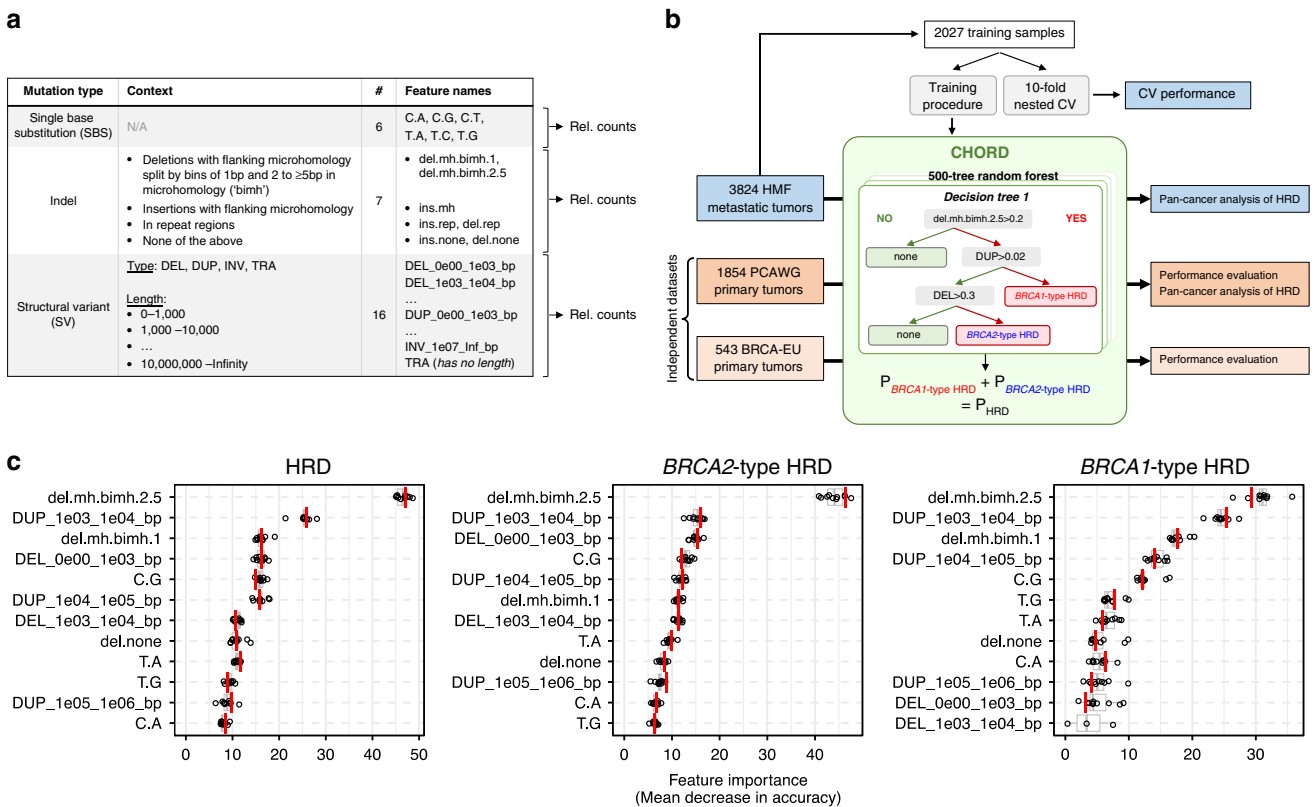

**Fig. 1 CHORD is a random forest Classifier of HOmologous Recombination Deficiency able to distinguish between *BRCA1*- and *BRCA2*-type HRD phenotypes in a pan-cancer context.** **a** The features used for training CHORD are relative counts of different mutation contexts, which fall into one of three groups based on mutation type. (i) Single nucleotide variants (SNV): six possible base substitutions (C > A, C > G, C > T, T > A, T > C, T > G). (ii) Indels: indels with flanking microhomology (del.mh, ins.mh), within repeat regions (del.rep, del.none), or not falling into either of these 2 categories (del.none, ins. none). (iii) Structural variants (SV): SVs stratified by type and length. Relative counts were calculated separately for each of the 3 mutation types. **b** Training and application of CHORD. From a total of 3,824 metastatic tumor samples, 2026 samples were selected for training CHORD. The model outputs the probability of *BRCA1*-type HRD and *BRCA2*-type HRD, with the probability of HRD being the sum of these 2 probabilities. The performance of CHORD was assessed via a 10-fold nested cross-validation (CV) procedure on the training samples, as well as by applying the model to the BRCA-EU dataset (543 primary breast tumors) and PCAWG dataset (1,854 primary tumors). Lastly, CHORD was applied to all samples in the HMF and PCAWG dataset in order to gain insights into the pan-cancer landscape of HRD. **c** The features used by CHORD to predict HRD as well as *BRCA1*-type HRD and *BRCA2*-type HRD, with their importance indicated by mean decrease in accuracy. Deletions with 2 to ≥5 bp (i.e. ≥2 bp) of flanking microhomology (del.mh. bimh.2.5) was the most important feature for predicting HRD as a whole, with 1–100 kb structural duplications (DUP_1e03_1e04_bp, DUP_1e04_1e05_bp) differentiating *BRCA1*-type HRD from *BRCA2*-type HRD. Boxplot and dots (n = 10) show the feature importance over 10-folds of nested CV on the training set, with the red line showing the feature importance in the final CHORD model. Boxes show the interquartile range (IQR) and whiskers show the largest/smallest values within 1.5 times the IQR.

genetic annotations (Fig. 2a). The concordance between predictions and annotations was quantified by calculating the area under the curve of receiver operating characteristic (AUROC) and precision-recall (AUPRC) curves (Fig. 2b, c). CHORD achieved excellent performance as shown by the high AUROC and AUPRC for all prediction classes (0.98 and 0.87, respectively). Additionally, CHORD achieved a maximum F1-score (~0.88) for predicting HRD at a cutoff of 0.5 which was thus set to be the classification threshold (Supplementary Fig. 6).

In the second approach, performance was evaluated on two independent datasets: the BRCA-EU dataset[8] (543 primary breast tumors) and the PCAWG dataset[14] (1854 primary tumors, pan-cancer). For both datasets, samples that (i) passed CHORD's QC criteria (i.e. MSI absent, ≥50 indels, ≥30 SVs if a sample was predicted HRD; Supplementary Notes and Supplementary Figs. 26 and 27) and (ii) for which the biallelic status of *BRCA1/2* could confidently be determined were selected for validation of CHORD. For the BRCA-EU dataset, this included the 365 samples that were used to train and evaluate the performance of

HRDetect[9]. For the PCAWG dataset, this included 1172 samples for which the same genetic criteria used for selecting samples from the HMF dataset for training CHORD applied. Applying CHORD on these samples revealed that the HRD probabilities were concordant to their *BRCA1/2* genetic status for both the BRCA-EU and PCAWG datasets (Fig. 2d, g). The AUROC (>0.98) and AUPRC (>0.93) values were comparable to those obtained by CV on the HMF training data for all prediction classes for both datasets (Fig. 2e, f, h, i). In the BRCA-EU dataset, we still observed some *BRCA1* deficient samples classified as HRP by CHORD (while HRDetect classified these as HRD) and tested whether this was due to differences in somatic calling algorithms. Indeed, using the variants obtained from the native pipeline of the HMF dataset (HMF pipeline[13]) for HRD prediction resulted in overall higher HRD probabilities compared to using the variants downloaded from ICGC, especially for *BRCA1*-deficient samples. This was apparent for sample PD4017 which became HRD using HMF pipeline called mutation profiles, with PD24186, PD11750, and PD23578 having greatly increased HRD probabilities

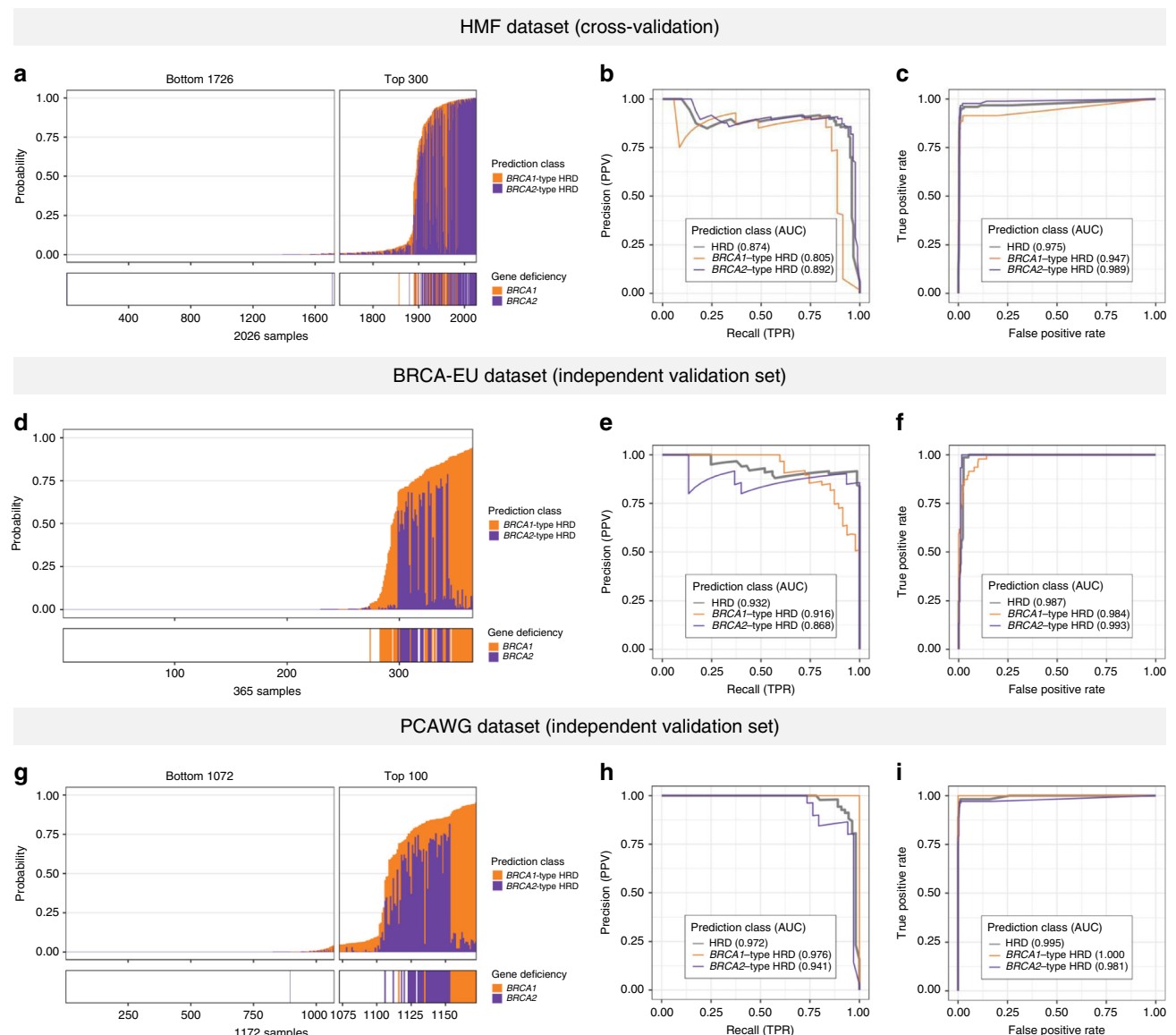

**Fig. 2 Performance of CHORD.** Performance was determined by 10-fold cross-validation (CV) on the HMF training data, as well as prediction on two independent datasets: BRCA-EU (primary breast cancer dataset) and PCAWG (primary pan-cancer dataset). BRCA-EU and PCAWG samples shown here all passed CHORD's QC criteria (i.e. MSI absent, ≥50 indels, ≥30 SVs if a sample was predicted HRD). **a**, **d**, **g** The probability of HRD for each sample (total bar height) with each bar being divided into segments indicating the probability of *BRCA1*- (orange) and *BRCA2*-type HRD (purple). Stripes below the bar plot indicate biallelic loss of *BRCA1* or *BRCA2*. In **a**, probabilities have been aggregated from the 10 CV folds. **b**, **e**, **h** Receiver operating characteristic (ROC) and **c**, **f**, **i** precision-recall curves (PR) and respective area under the curve (AUC) values showing the performance of CHORD when predicting HRD as a whole (gray), *BRCA1*-type HRD (orange), or *BRCA2*-type HRD (purple).

(Supplementary Fig. 7). Our results thus demonstrate that CHORD is robust when applied to other datasets. However, differences in variant calling pipelines can affect CHORD's ability to predict HRD (especially considering the still existing challenges of indel and SV calling from WGS data, and CHORD's dependency on these features). Additional validation and threshold optimization is thus recommended when applying CHORD on data from other variant calling pipelines.

The *BRCA1/2* deficient samples in the training set of CHORD primarily consisted of ovarian, breast, and prostate tumors, which could potentially bias CHORD's predictions if the mutation footprint of HRD is not universal across tissue types. We performed clustering of HMF and PCAWG samples using the input features for CHORD which revealed a cluster in which

the majority of samples predicted HRD by CHORD resided (Supplementary Fig. 8), suggesting that HRD mutational footprint is not tissue type specific. Next, to test whether CHORD is generalizable to all tissue types, we held out samples belonging to each cancer type from the training set (but grouped cancer types with few *BRCA1/2* deficient samples), and trained random forests in the same manner as was done for CHORD. These models were then applied to the held out HMF samples as well as PCAWG samples to calculate the likely prediction error for each cancer type (Supplementary Fig. 9). Using a classification cutoff of 0.5, we observed overall a low false positive rate (<2%) and false-negative rate (<6%). The false-negative rate was higher in biliary, lung and other cancer types, although these error estimates may not be entirely accurate due to the low number of

BRCA1/2 deficient samples in these cancer types. Our results indicate that CHORD likely has minimal cancer type bias.

We note that CHORD performs similarly to HRDetect based on predictions on the BRCA-EU dataset (AUROC = 0.98 for both models)[9]. In addition, the predictions of CHORD and HRDetect on the PCAWG dataset[10] were concordant for the vast majority of samples (1506/1526; 99%) (Supplementary Fig. 10). Of the 8 HRD samples only detected by CHORD, 3 showed biallelic loss of BRCA1/2, while none of the HRDetect-only samples could be explained by genetic biallelic loss. Given that CHORD, unlike HRDetect, does not rely on COSMIC SNV signatures[6] and SV signatures[8], the similar performance between the two models suggests that accurate detection of HRD is possible without using an intermediate mutational signature extraction step[15]. To further validate this, we trained a random forest model (CHORD-signature) that uses the SNV and SV signatures as input instead of mutation contexts. CHORD-signature performed similarly to CHORD (Supplementary Fig. 12), which can be explained by the reliance on similar features (Supplementary Fig. 11), namely microhomology deletions and SV signature 3 (analogous to 1–100 kb duplications). We thus conclude that accurate detection of HRD does not require mutational signatures, thereby simplifying HRD calling and avoiding potential complications associated with the fitting step required for computing signature contributions in individual samples (for which there is currently no consensus approach)[15].

**Effect of treatment on HRD predictions**. The HMF dataset comprises tumors from patients with metastatic cancer who have been exposed (some heavily) to treatment which could potentially affect CHORD's predictions. Two recent studies showed that common cancer treatments in general do not induce mutations that may interfere with CHORD predictions[16,17]. However, these two studies (as well as one by Behjati et al.[18]) did show that radiotherapy had the potential to induce deletions with flanking microhomology, which could potentially lead to false-positive HRD classifications. To investigate this, we used random forests to identify and compare the mutational features associated with radiotherapy and BRCA1/2 deficiency when using clonal variants versus subclonal variants (which are enriched for treatment induced mutations[16,19]) as input features. This revealed that small deletions with 1 bp of flanking homology (del.mh.bimh.1) are highly associated with radiotherapy (Supplementary Fig. 14) and less with BRCA1/2 deficiency. When we retrained CHORD with all microhomology deletions merged into a single feature (CHORD-del.mh.merged; Supplementary Fig. 15), there were only few discrepant predictions (9 CHORD-specific and 5 CHORD-del.mh.merged-specific out of 3715 samples; Supplementary Fig. 16). All 5 samples that were CHORD-del.mh.merged-specifc did have radiotherapy as a previous treatment, while of the 9 samples predicted HRD only by CHORD, 5 had radiotherapy although 2 had evidence of BRCA1/2 biallelic loss. These data suggest that splitting microhomology deletions into two microhomology length bins may slightly reduce false positive predictions resulting from radiotherapy treatment, although the low number of discrepant samples between CHORD and CHORD-del.mh.merged also indicates that the impact of radiotherapy on HRD prediction is minimal, at least when using all somatic variants (clonal plus subclonal) as input (which is likely the default setting for routine application).

On the other hand, we observed more samples being predicted as HRD based on subclonal variants but HRP based on clonal variants for CHORD-del.mh.merged (97 samples) compared to CHORD (64 samples) (Supplementary Fig. 17a, b). This indicates that having microhomology deletions split by these two

homology length bins may mitigate false-positive predictions when CHORD is applied to subclonal variants, whether due to mutations induced by radiotherapy, other treatments, or noise from variant calling algorithms. Alternatively, some samples that are scored HRP based on clonal variants but HRD on subclonal variants could truly be HRD, especially since 4 of these samples had evidence of BRCA1/2 biallelic loss (deep deletion: $n = 1$; LOH and a pathogenic variant: $n = 1$; 2 pathogenic variants: $n = 2$). For these samples, it is likely that BRCA1/2 biallelic loss occurred relatively late in the tumor progression stage which results in an insufficient number of HRD-associated mutations for clear HRD classification by CHORD. Furthermore, subclonal-only HRD could potentially also be explained by transient inactivation of HR e.g. through epigenetic silencing of key components. Thus, CHORD predictions on subclonal variants must be interpreted with caution, especially given the extra challenges associated with accurately detecting subclonal variants with low variant allele frequency (VAF).

**BRCA2, RAD51C, and PALB2 are associated with BRCA2-type HRD while only BRCA1 is associated with BRCA1-type HRD**. To gain insights into the genetic causes of HRD, we applied CHORD to both the HMF and PCAWG datasets and selected the samples that passed CHORD's QC criteria (i.e. MSI absent, ≥50 indels, ≥30 SVs if a sample was predicted HRD; Supplementary Notes). For the HMF dataset, we also selected a single tumor per patient (based on highest tumor purity) for those with multiple biopsies, though all patients had consistent HRD probabilities across all biopsies (Supplementary Data 1). This yielded a total of 5122 patients (3504 from HMF and 1618 from PCAWG), with 310 (6%) being classified as being homologous recombination deficient (CHORD-HRD). Of these, 118 were classified as having BRCA1-type HRD and 192 as having BRCA2-type HRD. The remaining 4,812 patients were classified as homologous recombination proficient (CHORD-HRP) (Fig. 3a and Supplementary Data 1).

We then sought to identify the key mutated genes underlying the HRD phenotype by performing an enrichment analysis of biallelically inactivated genes in CHORD-HRD vs. CHORD-HRP patients. For this analysis, we started from a list of 781 genes that are cancer related (based on the catalog of genes from Cancer Genome Interpreter) and/or HR related (manually curated based on the KEGG HR pathway, as well as via literature search) (Supplementary Data 3). For these genes, we considered likely pathogenic variants (according to ClinVar) as well as predicted impactful variants such as nonsense mutations to contribute to gene inactivation (see "Methods"). This revealed that, in addition to BRCA1 and BRCA2 ($q < 10^{-5}$ for both genes, one sided Fisher's exact test), RAD51C and PALB2 ($q < 0.001$ and $q < 0.05$ respectively) were also significantly enriched amongst HRD patients using a q-value threshold of 0.05 (Fig. 3b).

Of all CHORD-HRD HMF patients, ~60% (184/310) could be explained by biallelic inactivation of either BRCA2 (cluster 1; $n = 117$), BRCA1 (cluster 5; $n = 54$), RAD51C (cluster 2; $n = 6$), or PALB2 (cluster 3; $n = 7$), which was most often caused by LOH in combination with a pathogenic variant or frameshift, or a deep deletion (Fig. 3c and Supplementary Data 4). RAD51C and PALB2 were recently linked to HRD as incidental cases using mutational signature-based approaches[11,20] and our results now confirm that biallelic inactivation of these two genes results in HRD and is actually a common cause of HRD (albeit to a lesser extent than for BRCA1/2). RAD51C and PALB2 deficient patients shared the BRCA2-type HRD phenotype (absence of duplications) with BRCA2 deficient patients (clusters 1–3; Fig. 3c), consistent with previous studies[11,12]. On the other hand, only

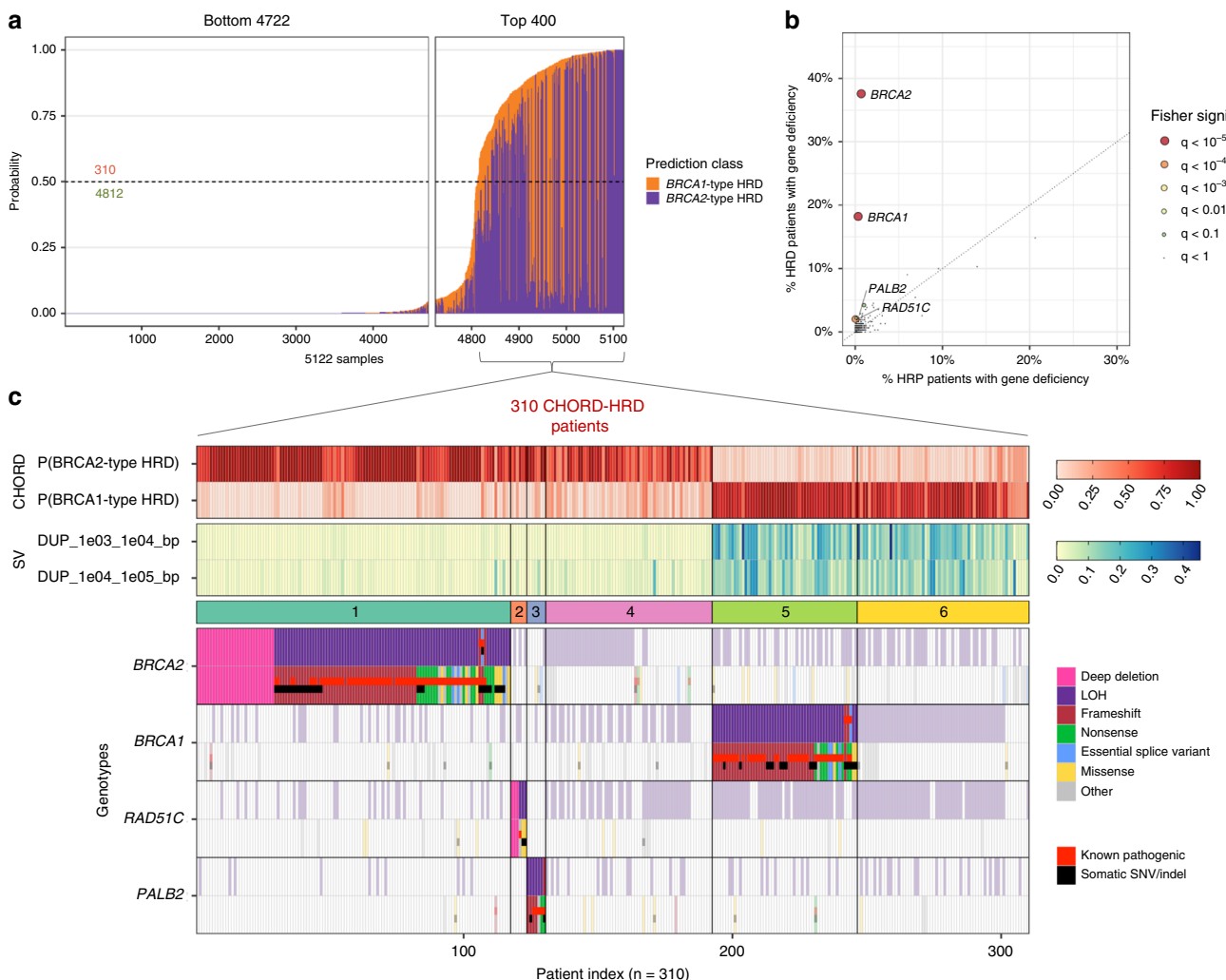

**Fig. 3 The genetic causes of HRD in patients from the HMF and PCAWG datasets. a** The bar plot shows the probability of HRD for each patient (total bar height) with each bar being divided into segments indicating the probability of *BRCA1*-type HRD (orange) and *BRCA2*-type HRD (purple). 310 patients were predicted HRD while 4812 were predicted HRP by CHORD. **b** A one-tailed Fisher's exact test identified enrichment of *BRCA1* ($q = 9.4e-51$), *BRCA2* ($q = 4.8e-101$), *RAD51C* ($q = 5.6e-5$) and *PALB2* ($q = 0.02$) biallelic inactivation in CHORD-HRD vs. CHORD-HRP patients (from a list of 781 cancer and HR related genes). Each point represents a gene with its size/color corresponding to the statistical significance as determined by the Fisher's exact test, with axes indicating the percentage of patients (within either the CHORD-HRD or CHORD-HRP group) in which biallelic inactivation was detected. Multiple testing correction was performed using the Hochberg procedure. **c** Biallelic inactivation of *BRCA2*, *RAD51C* and *PALB2* was associated with *BRCA2*-type HRD, whereas only *BRCA1* inactivation was associated with *BRCA1*-type HRD. Top: *BRCA1*- and *BRCA2*-type HRD probabilities from CHORD. Middle: SV contexts (duplications 1–10 kb and 10–100 kb) used by CHORD to distinguish *BRCA1*- from *BRCA2*-type HRD. Bottom: The biallelic status of each gene. Samples were clustered according to HRD subtype, and by the impact of a biallelic/monoallelic event (based on 'P-scores' as detailed in the methods). Clusters 1, 2, 3, and 5 correspond to patients with identified inactivation of *BRCA2*, *RAD51C*, *PALB2* and *BRCA1*, while clusters 4 and 6 correspond to patients without clear biallelic inactivation of these 4 genes. Tiles marked as "Known pathogenic" refer to variants having a "pathogenic" or "likely pathogenic" annotation in ClinVar. "Other" variants include various low impact variants such as splice region variants or intron variants (these are fully specified in Supplementary Data 4). LOH: loss-of-heterozygosity. Only data from samples that passed CHORD's QC criteria are shown in this figure (MSI absent, ≥50 indels, and ≥30 SVs if a sample was predicted HRD).

*BRCA1* deficient patients (cluster 5) harbored the *BRCA1*-type HRD phenotype (1–100 kb duplications).

Of note, we observed one patient (Fig. 3c; patient #6) bearing a known pathogenic frameshift mutation in *BRCA1* (Supplementary Data 4; patient HMF001925, c.1961dupA), which based on current practices for detecting HRD in the clinic (testing for pathogenic SNVs/indels)[5] would be considered the driver mutation. However, our genetic analysis indicates that the deep deletion in *BRCA2* (which would be missed by testing for SNVs/indels) was the cause of HRD, which is supported by the lack of LOH in *BRCA1*, as well as the *BRCA2*-type HRD phenotype of this patient.

In ~40% of CHORD-HRD patients (126/310; clusters 4 and 6, Fig. 3c), there was no clear indication of biallelic loss of *BRCA1/2*, *RAD51C* or *PALB2* (henceforth referred to as the "HRD associated genes"). However, 109 of these patients had a deleterious event in a single allele of one of the HRD associated genes (the majority due to LOH (including copy number neutral LOH)), with a similar cancer type distribution in these patients as in the biallelically affected patients (Supplementary Fig. 21). Some samples had, as a second hit, variants not known to be pathogenic, but could potentially be novel pathogenic variants (Supplementary Notes and Supplementary Fig. 28 and Supplementary Data 7). We also found enrichment of LOH in *BRCA1*,

*BRCA2*, as well as *RAD51C* in HRD samples (Supplementary Fig. 22), which implies the involvement of LOH in the inactivation of these genes for the patients in clusters 4 and 6. This is consistent with the finding by Jonsson et al.[21] that LOH is enriched in tumors with *BRCA1/2* germline pathogenic variants or somatic loss-of-function variants. Davies et al.[9] showed that promoter methylation of *BRCA1* was present in 22% of ovarian and 16% of breast primary cancers with HRD (Supplementary Table 1). *BRCA1* and *RAD51C* promoter methylation with loss of the other allele was also reported in HRD tumors in other studies[11,12,20]. Thus, *BRCA1* and *RAD51C* promoter methylation, likely in combination with LOH, may have led to the HRD phenotype for a sizable portion of the ovarian and of breast cancer patients with no clear biallelic loss of the HRD associated genes, and potentially for patients with other cancer types as well (Supplementary Fig. 23). Unfortunately, we could not directly assess this as methylation data was not available for the HMF nor the PCAWG dataset.

We also cannot rule out the possibility that deficiencies in other HR genes that did not reach significance in our enrichment analysis, underlie the HRD phenotype for a small number of patients in clusters 4 and 6. We indeed identified 17 patients with biallelic inactivation of a HR gene other than *BRCA1/2*, *RAD51C* or *PALB2*, and 1 patient with a likely inactivating biallelic event (LOH in combination with a nonsense variant in *CHEK1*) (Supplementary fig. 24). Notably, the 4 patients with *RAD51B* (n = 2) and *XRCC2* (n = 2) deficiency were all predicted to have *BRCA2*-type HRD, a phenotype shared with *RAD51C* deficient patients[22]. Given that these three genes all belong to the RAD51 paralog complex BCDX2[23], the *BRCA2*-type HRD suggests that *RAD51B* and *XRCC2* deficiency could have led to HRD in these patients. Likewise, the 4 patients with deficiencies in the *BRCA1*-binding proteins, *BARD1*[24] (n = 1), *BRIP1*[25] (n = 1), *FAM175A*[26] (n = 1) and *FANCA*[27] (n = 1), were all predicted as having *BRCA1*-type HRD. Thus, while we could not conclusively determine the cause of HRD for patients in clusters 4 and 6, we postulate that HRD in these patients may have been a result of epigenetic silencing of *BRCA1/2* or *RAD51C*, deficiencies in other HR genes (not associated to HRD in our analysis), or possibly a result of other unknown regulatory mechanisms.

**The incidence and genetic cause of HRD varies in different tissue types and cancer stage.** We next investigated the differences in the incidence and genetic causes of HRD based on primary tumor location in both primary (PCAWG) and metastatic (HMF) cancer datasets (Fig. 4). HRD was most prevalent in ovarian, breast, prostate and pancreatic cancer (85% combined), and only occurred sporadically in other cancer types (15%) (Supplementary Data 5). Compared to metastatic cancer, HRD is found much more often in primary ovarian (52% vs 30%) and breast (24% vs 12%) cancers, and less often in primary prostate (5.6% vs 13%) and pancreatic (7.3% vs 13%) cancer (Fig. 4a). Notably, in metastatic cancer, prostate and pancreatic cancer have a similar incidence of HRD to breast cancer (all ~13%). However, the observed differences in HRD rates between the primary and metastatic cohorts may not necessarily be conclusive as we can not rule out confounding factors such as patient inclusion criteria.

Across different cancer types, we observed pronounced diversity in HR function loss (Fig. 4b). *BRCA2*-type HRD deficiencies (including *BRCA2*, *RAD51C*, *PALB2* deficiencies) were more frequent in pancreatic and prostate cancer. On the other hand, *BRCA1*-type HRD deficiencies were found more often in ovarian and breast cancer. Interestingly, for ovarian and prostate cancer, *BRCA1*-type HRD deficiencies were more prominent in primary cancer compared to metastatic cancer.

Whether these differences in gene deficiencies in different cancer types can be linked to a biological cause or have prognostic value remains to be determined.

In 94% (292/310) of all CHORD-HRD patients, we found mono- or biallelic inactivation of at least one of the four HRD associated genes (*BRCA1*, *BRCA2*, *PALB2*, *RAD51C*; Fig. 4c). In the case of biallelic inactivation, we observed LOH to be the dominant secondary event, occurring in combination with a germline SNV/indel (33%) or with a somatic SNV/indel (14%) of CHORD-HRD patients. LOH of *BRCA1/2*, *RAD51C* or *PALB2* was also found as a monoallelic event, mainly in ovarian (47%) and breast (49%) cancer patients (Supplementary Data 5). As indicated earlier, the other allele may be inactivated by epigenetic mechanisms in these patients (or alternatively HRD was caused by inactivation of another HR gene). Interestingly, we find that deep somatic deletions do frequently contribute to biallelic loss of *BRCA2* or *RAD51C*, occurring in 10% of CHORD-HRD patients pan-cancer (Supplementary Data 5). However, deep deletions (primarily of *BRCA2*; Supplementary Fig. 23) occurred much more frequently in prostate cancer (33%) compared to other cancer types, consistent with previous observations[28]. Nevertheless, deep deletions of HRD genes did occur in every cancer type with a high frequency of CHORD-HRD patients indicating that complete somatic gene loss is a common and underestimated cause of HRD in both primary and metastatic cancer.

We find that biallelic gene loss is often associated with germline predisposition (Fig. 4d) in ovarian (32%), breast (36%), and pancreatic (56%) cancer patients, but to a lesser extent in prostate cancer patients (24%) (Supplementary Data 5). On the other hand, biallelic gene loss exclusively by somatic events occurs in sizable proportion of CHORD-HRD patients (35% pan-cancer), being most frequent in prostate cancer (54%) (Supplementary Data 5) mainly due to the deep deletions (Supplementary Fig. 23). Although these frequencies may not be fully representative for each cancer type due to the proportion of patients with unknown mutation status in at least one allele (indicated as "Unknown" in Fig. 4d), these observations do emphasize that somatic-only events should not be overlooked as a mechanism of HR gene inactivation.

## Discussion

Here we describe a classifier (CHORD) that can detect HRD (as well as HRD sub-phenotypes) across cancer types based on mutation profiles. By using this tool in systematic pan-cancer analysis, we reveal novel insights into the mechanisms and incidence of HRD across cancer types with potentially important clinical relevance.

HRD targeted therapy with PARPi is mostly restricted to breast and ovarian cancer[5], though its use for treating pancreatic cancer was recently approved by the FDA (US Food and Drug Administration)[29]. However, we show that HRD is common not only in ovarian and breast cancer, but also in prostate and pancreatic cancer. The incidence of HRD was relatively higher in metastatic prostate and pancreatic cancer, and lower for ovarian and breast cancer as compared with primary tumors. This may reflect more intensive familial (germline) testing for *BRCA1/2* mutations in ovarian and breast cancer[30] and consequently earlier diagnosis and treatment with fewer cases of progression to metastatic cancer as a result. However, we cannot formally exclude that these observations originate from differences in cohort inclusion criteria that could skew numbers (e.g. due to more recruitment of patients with triple negative breast cancer which has higher HRD rates[12]).

We show that HRD is also found sporadically in cancer types other than breast, ovarian, prostate or pancreatic, but collectively

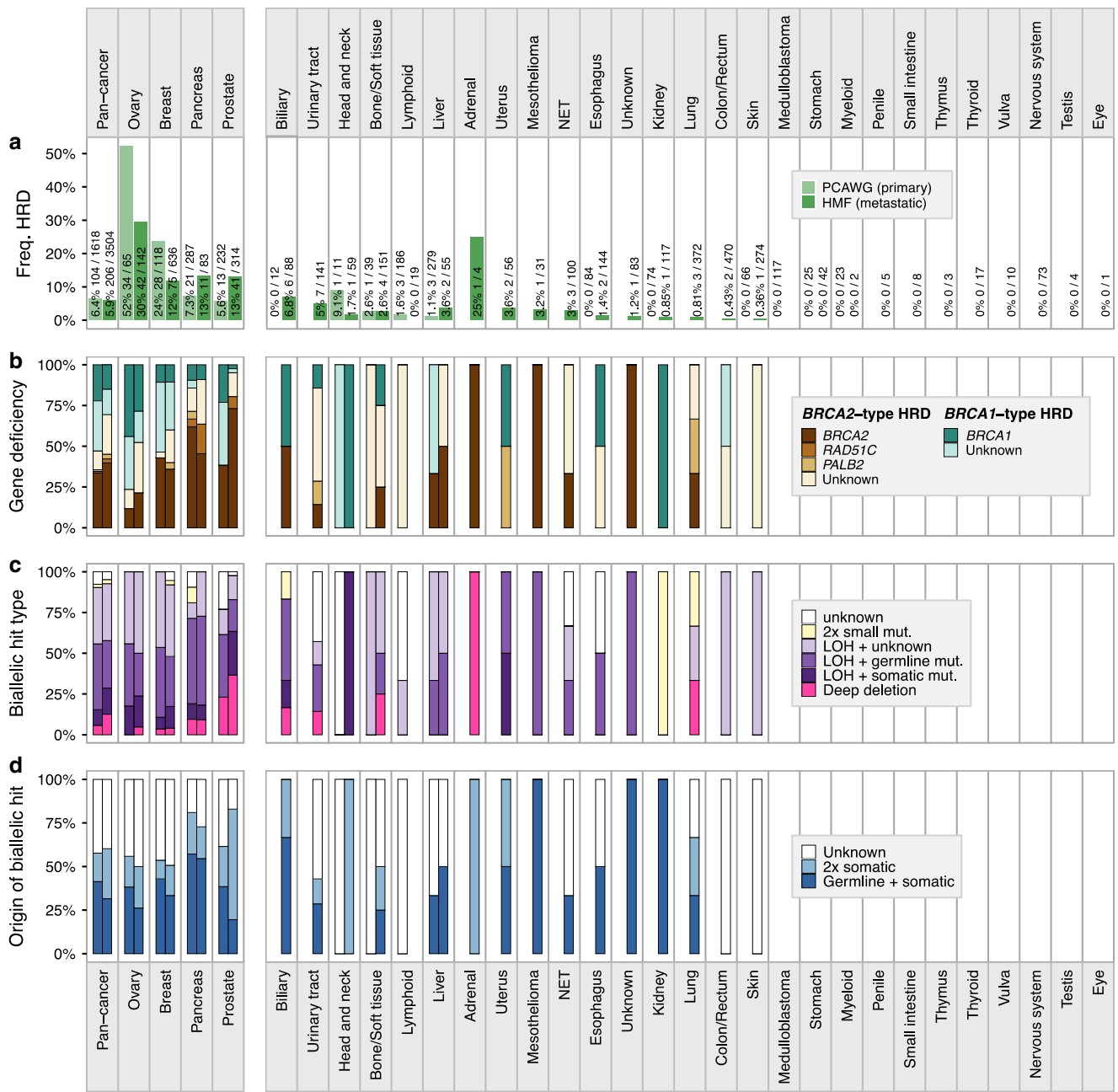

**Fig. 4 Percentage breakdown of the incidence and genetic causes of HRD in CHORD-HRD patients pan-cancer and by cancer type.** Left and right bars represent the HMF and PCAWG datasets respectively. The vertical split in the figure separates cancer types with (left side) and without (right side) ≥10 CHORD-HRD patients in at least one of the datasets. **a** Frequency of HRD. Cancer types where no frequency of HRD is displayed contain no data in either the HMF or PCAWG datasets. **b** The gene deficiency associated with HRD. Bar segments are grouped into *BRCA2*-type HRD genes (*BRCA2*, *RAD51C*, *PALB2*) and *BRCA1*-type HRD genes (*BRCA1* only). **c** The likely combination of biallelic events in *BRCA1/2*, *RAD51C* or *PALB2* causing HRD. **d** Whether the genetic cause of HRD was purely due to somatic events, due to germline predisposition, or unknown. In **c**, **d**, "Unknown" and/or "LOH + unknown" bar segments refer to patients where no clear biallelic loss of the aforementioned *BRCA1/2*, *RAD51C*, or *PALB2* was identified (i.e. clusters 4 and 6 of Fig. 3c). LOH: loss-of-heterozygosity. Only data from samples that passed CHORD's QC criteria are shown in this figure (MSI absent, ≥50 indels, and ≥30 SVs if a sample was predicted HRD).

this constitutes a sizable group of patients (15% of all patients). We do acknowledge that there may be underestimation of HRD frequency in these other cancer types due to the low prevalence of BRCA1/2 deficient samples, which served as examples of HRD samples for training CHORD (Supplementary Fig. 23). On the other hand, we have shown that the HRD mutational footprint is not tissue type specific (Supplementary Fig. 8) suggesting that

cancer type biases in the training set should not impact CHORD predictions. Our results thus indicate that a large number of patients who would potentially benefit from PARPi therapy still remain unnoticed. Since the mutational phenotype of HRD is independent of cancer type, mutational scar based HRD detection such as with CHORD would be valuable for cancer type agnostic patient stratification for future PARPi trials[31]. This is particularly

important for metastatic patients (who depend on systemic treatments and benefit most from targeted treatments like PARPi), as well as for cancer types currently lacking good markers for patient stratification for such treatment (such as prostate[32] and biliary[33] cancer).

Genetic based detection of HRD in the clinic is commonly done by testing for pathogenic *BRCA1/2* germline mutations[5]. However, such hereditary mutations are only present in 30% of CHORD-HRD patients (Supplementary Notes and Supplementary Fig. 29) indicating that germline testing likely misses a substantial number of HRD patients. Germline variant testing is particularly unsuitable for prostate cancer where gene inactivation is frequently caused by somatic deep deletions, which prevent the identification of any SNVs/indels at the affected locus when using panel- or PCR-based sequencing methods (exon scanning). This problem also exists for other cancer types where deep deletions also make up a non-negligible fraction of HR gene inactivation cases. While somatic mutation testing improves diagnostic yield and is indeed increasingly performed in the clinic[5], WGS based genetic testing is ultimately necessary to capture the full spectrum of genetic alterations and to accurately determine the mutational status of HR genes. However, even such broad genetic testing with focus on biallelic gene inactivation still potentially misses roughly 50% of all HRD patients (Supplementary Notes and Supplementary Fig. 29).

We do acknowledge that mutational scars represent genomic history and not current on-going mutational processes that can result in false positive CHORD predictions, which could be for example due to reversion of HRD by secondary frameshifts[34,35], or recent acquisition of the HRD phenotype. False-positive predictions could also arise from treatments producing similar mutational scars (in particular microhomology deletions) to HRD. The most common cancer treatments have been shown to have little or no contribution to microhomology deletions, with the exception of radiotherapy[16–18]. However, we showed that radiotherapy itself likely does not lead to false-positive predictions. We cannot exclude the possibility however that clonal expansion of a radiotherapy resistant tumor cell leads to sufficient enrichment of radiotherapy associated microhomology deletions in the tumor, resulting in a false positive prediction. Ultimately, the ability for CHORD to improve patient stratification and treatment outcome will need to be evaluated in direct comparisons and prospective clinical trials.

Thus, while CHORD can detect HRD independent of the underlying cause, genetic testing of HRD genes is complementary and can provide supporting information for making a final verdict on a patient's HR status. The unique advantage of using WGS, although not routine in clinical diagnostics yet, but likely in the near future[36], is that both genetic testing and mutational scar based HRD detection with CHORD can be performed simultaneously with the same assay. We envision that the findings from our analyses incentivizes improvements to current clinical practices for detecting HRD, and that the application of genomics-based approaches, like CHORD, in the clinic will support these endeavors and provide additional treatment options for patients. CHORD is freely available as an R package at https://github.com/UMCUGenetics/CHORD.

## Methods

**Datasets.** We have used patient data for which re-use for cancer research was consented by the patients as part of two clinical studies (NCT01855477, NCT02925234) unrelated to the current work. Matched tumor/blood samples from these patients were sequenced and uniformly analyzed by the Hartwig Medical Foundation (HMF; https://www.hartwigmedicalfoundation.nl/en/appyling-for-data/). The data transfer agreement (Data Request 10 and 47) were approved by the medical ethical committees (METC) of the University Medical Center Utrecht. We received germline and somatic VCF files of the 3,824 metastatic tumor samples from 3,584 patients in May 2019. For patients with multiple biopsies that were taken at different timepoints, patient IDs were suffixed by "A" for the first biopsy, "B" for the second biopsy, etc (e.g. HMF001423A, HMF001423B). A detailed description of the whole patient cohort has been described in detail in Priestley et al.[13].

Somatic variant TSV files of the 560 breast cancer (BRCA-EU) dataset were downloaded from the International Cancer Genome Consortium (ICGC; https://dcc.icgc.org/repositories) in August 2017. BAM files for the 44 BRCA-EU samples are available from EGA (datasets: EGAD0001000063, EGAD00001001322, EGAD00001001337). *BRCA1/2* status annotations for this dataset being obtained from the supplementary data in Davies et al.[9].

Somatic variant VCF files and somatic copy-number TSV files for the ICGC portion of the Pan-Cancer Analysis of Whole Genomes (PCAWG) dataset (consisting of 1854 patient tumors) were downloaded from https://dcc.icgc.org/releases/PCAWG on 3 March 2020. PCAWG access for germline data has been granted via the Data Access Compliance Office (DACO) Application Number DACO-1050905 on 6 October 2017 and via https://console.cancercollaboratory.org download portal on 4 December 2017. Germline VCF files were downloaded from the cancer collaboratory download portal on 21 March 2018.

**Variant calling.** Variant calling in the HMF dataset was performed previously by HMF (https://github.com/hartwigmedical/pipeline)[13]. Briefly, reads were mapped to GRCh37 using BWA-MEM v0.7.5a with duplicates being marked for filtering. Indels were realigned using GATK v3.4.46 IndelRealigner. GATK Haplotype Caller v3.4.46 was used for calling germline variants in the reference sample. For somatic SNV and indel variant calling, GATK BQSR3 was first used to recalibrate base qualities, followed by Strelka v1.0.14 for the variant calling itself. Somatic SV calling was performed using GRIDSS v1.8.0. Copy-number calling was performed using PURity & PLoidy Estimator (PURPLE), that combines B-allele frequency (BAF), read depth, and structural variants to estimate the purity and copy number profile of a tumor sample[37] as well as VAF and clonality (either clonal, subclonal or inaccurate) estimates of each variant.

**Determining gene biallelic status.** For samples in the HMF and PCAWG cohorts, biallelic status was determined for 781 genes (Supplementary Data 3) which included genes associated with cancer, according to Cancer Genome Interpreter (https://www.cancergenomeinterpreter.org/genes), as well as a manually curated set of genes involved in HR (based on the KEGG HR pathway (https://www.genome.jp/), as well as via a literature search. This was performed using an in-house pipeline that interprets copy-number, and germline and somatic SNV/indel data from the HMF variant calling pipeline to determine biallelic gene status (https://github.com/UMCUGenetics/hmfGeneAnnotation).

First, the copy number status in the gene region was determined. If the minimum copy number was <0.3, the gene was considered to have a deep deletion (and by default biallelically inactivated). Else, the gene was screened for 2 mutation events, which included following combinations: (i) loss-of-heterozygosity (LOH) with a germline or somatic SNV/indel; (ii) a germline and somatic SNV/indel; or (iii) 2 somatic SNV/indels.

LOH was considered pathogenic and was automatically given a pathogenicity score (*P*-score) of 5. LOH occurred if the minimum minor allele copy number within a gene region was <0.2.

Pathogenicity of SNVs/indels was assessed based on pathogenicity annotations from ClinVar (https://www.ncbi.nlm.nih.gov/clinvar/; GRCh37, database date 2020-02-24). For variants without an entry in ClinVar, pathogenicity was assessed based on variant type as determined by SnpEff (http://snpeff.sourceforge.net/; v4.1 h). Briefly, variants can be given one of the following annotations from ClinVar: pathogenic, likely pathogenic, variant of unknown significance (VUS), likely benign, and benign. A *P*-score of 1–5 was also assigned to each annotation, with 1 = benign and 5 = pathogenic. In addition, variant types as determined by SnpEff were assigned similar annotations and scores: out-of-frame frameshifts were considered pathogenic (*P*-score = 5); nonsense and splice variants were considered likely pathogenic (*P*-score = 4); missense variants, essential splice variants, and inframe frameshifts were considered VUS's (*P*-score = 3); the remaining variant types were considered likely benign or benign (*P*-score ≤ 2). The final *P*-score of a variant was the ClinVar *P*-score if a ClinVar annotation exists for that variant, and if not, the SnpEff *P*-score was used. See Supplementary Data 6 for details on pathogenicity scoring.

*P*-scores from pairs of mutation events (i.e. SNV, indel, or LOH) were summed to yield a biallelic pathogenicity score (*BP*-score), giving a maximum possible score of 10. Deep deletions were automatically given a score of 10. Per gene, the biallelic event with the highest score was taken the biallelic status of the gene. If multiple events had the same score, a biallelic event was greedily selected.

**Extracting mutation contexts.** The counts of three types of mutation contexts (SNV, indel, and structural variant (SV) contexts) were determined from the somatic variant data from the HMF, PCAWG and BRCA-EU cohorts

(Supplementary Data 2). This was performed using the R package *mutSigExtractor* (https://github.com/UMCUGenetics/mutSigExtractor).

The SNV contexts comprised of 96 trinucleotide contexts, which are composed of one of six classes of base substitutions (C > A, C > G, C > T, T > A, T > C, T > G) in combination with the immediate 5′ and 3′ flanking nucleotides.

The indel contexts comprised of 6 types based on the presence of: short tandem repeats (ins.rep, del.rep); short stretches of identical sequence at the breakpoints, also known as microhomology (ins.mh, del.mh); or the presence of neither (ins. none, del.none). Indels in repeat regions were defined as the presence of ≥1 copy of the indel sequence downstream (i.e. in the 3′ direction) from the breakpoint, where sequence length must be <50 bp. Indels with flanking microhomology were defined as the presence of the following sequence features up or downstream from the breakpoint: (i) ≥1 copy of the indel sequence if the indel sequence length is ≥50 bp; (ii) ≥2 bp sequence identity to the indel sequence; or (iii) ≥1 bp sequence identity if the indel sequence length is ≥3 bp. For (ii) and (iii) the number of up or downstream bases searched was equal to the length of the indel. The 6 indel contexts types were further expanded into 30 indel contexts by stratifying ins.rep, del.rep, ins.none, and del.none by indel sequence length (1–4 bp and ≥5 bp); and ins.mh and del.mh by the number of bases in microhomology ("bimh"; 1–4 bp and ≥5).

The 16 SV contexts were composed of the SV type (deletion, duplication, inversion, translocation) and the SV length (1–10 kb, 10–100 kb, 100kb–1Mb, 1–1Mb, >10 Mb). Note that SV length is not applicable for translocations.

## Random forest training

*Features.* To construct the features for training the CHORD, the 96 trinucleotide contexts were simplified to six base substitution contexts by discarding the 5′ and 3′ flanking nucleotide information. For CHORD-del.mh.merged, the 30 indel contexts were simplified to the 6 indel types. For CHORD and CHORD-signature, the del.mh indel type was split into 2 bins: del.mh with 1 bp homology and 2 to ≥5 (i.e. equivalent to ≥5 bp) homology (del.mh.bimh.1 and del.mh.bimh.2.5 respectively). Then, relative contribution was calculated for each feature per mutation context type (i.e. SNV, indel and SV contexts separately). For CHORD-signature, the 96 trinucleotide contexts were fitted to the 30 COSMIC SBS signatures[9] using the non-negative least squares algorithm (incorporated in *mutSigExtractor*). The SV contexts were fitted in the same manner to the 6 SV signatures[9]. The relative contribution of the SBS signatures, SV signatures, and indel contexts was then calculated per mutation type.

*Training set.* The training set consisted of samples which we could confidently consider *BRCA1/2* deficient or proficient based on the *P*-scores/*BP*-scores as described in Determining gene biallelic status and Supplementary Data 6. *BRCA1/2* deficiency was defined as having a *BP*-score = 10. This includes samples with: (i) a deep deletion, (ii) LOH in combination with a pathogenic SNV/indel or an out-of-frame frameshift, or (iii) two pathogenic SNV/indels and/or or out-of-frame frameshifts. Within the *BRCA1/2* deficient group, samples where the absolute frequency of indels within repeat regions was >14,000 were considered to have microsatellite instability (MSI) and were removed. This filtering step was done as the relative contribution of indels in repeat regions are grossly overrepresented in samples with MSI, thereby masking the contribution of microhomology deletions. This sample group ultimately consisted of 35 *BRCA1* ("*BRCA1*" class) and 89 *BRCA2* ("*BRCA2*" class) deficient samples which were both considered HRD during the training. Conversely, *BRCA1/2* proficiency required the following criteria: (i) Absence of deep deletions or LOH; (ii) all SNV/indels had a *P*-score ≤ 3 (VUS or lower in impact); (iii) for the highest impact pair of SNV/indels (i.e. highest *BP*-score), both variants had a *P*-score ≤ 3 (VUS or lower in impact). This *BRCA* proficient group ("*none*" class) consisted of 1902 samples which were considered HRP during the training (Supplementary Fig. 1).

*Training procedure.* The training procedure for CHORD (as well as other models described in this study) is illustrated in Supplementary fig. 2. A core training procedure, which performs feature selection and class resampling, forms the basis for the full training procedure (Supplementary Fig. 2a). Feature selection was done to retain mutation contexts which were significantly higher (*p* < 0.01, determined by one-tailed Wilcoxon tests) in *BRCA1/2* deficient versus proficient samples. Class resampling serves to reduce the difference in the number of samples between each class (i.e. class imbalances). Here, a grid search was performed to determine the optimal pair of the following parameters: (i) down-sampling of the "*none*" class: 1x (i.e. no down-sampling), 2x or 4x; (ii) up-sampling of the "*BRCA1*" class: 1x (i.e. no up-sampling), 1.5x or 2x. For each iteration of the grid search, 10-fold cross-validation (CV) was performed, after which the AUPRC was calculated. The parameter pair with the highest AUPRC was chosen. With the selected features and resampling parameters, a random forest was then trained that predicts the probability of a new sample being in one of the aforementioned three classes (i.e. "*BRCA1*", "*BRCA2*" or "*none*"). We defined the HRD probability as the sum of the probability of belonging to the "*BRCA1*" and "*BRCA2*" classes, where a sample was considered HRD if the HRD probability was ≥0.5. Random forests were trained using the *randomForest* R package.

The full training procedure was split into two stages (Supplementary Fig. 2b). The first stage serves to filter "*BRCA1*" or "*BRCA2*" samples from the which are

likely not HRD (e.g. due to reversal of biallelic inactivation via a second frameshift bringing the gene in frame), or "*none*" samples, which are likely not HRP (e.g. due to deficiencies in other HR genes). Here, the core training procedure is encapsulated by a 10-fold CV loop to allow every sample to be excluded from the training set to subsequently calculate an unbiased HRD probability. This was repeated 100 times and the number of times each sample was HRD or HRP was calculated. "*BRCA1*" or "*BRCA2*" samples that were predicted HRD < 60 times were blacklisted while "*none*" samples that were predicted HRD > 40 times were blacklisted. In the second training stage, the core training procedure was performed on a training set without the blacklisted samples. This yielded the final random forest model.

The performance of the final random forest model was assessed using two approaches: (i) 10-fold CV of the training set by further encapsulating the full training procedure in a 10-fold CV loop; (ii) applying the final random forest model to an external dataset (BRCA-EU dataset). An AUPRC was then calculated for both approaches. In the case of the BRCA-EU dataset, *BRCA1/2* deficiency annotations were obtained from Davies et al.[9]. All performance metrics were calculated using the *mltoolkit* R package (https://github.com/UMCUGenetics/mltoolkit).

**Determining the genetic cause of HRD**. To determine the genetic cause of HRD, tumors were first selected from the HMF cohort based on the absence of MSI, having ≥50 indels, and ≥30 SVs for HRD predicted samples (Supplementary Data 1). Furthermore, for patients with multiple biopsies, a single tumor per patient was selected (based on the one with highest tumor purity). In total, 3504 tumors were selected (from the 3824 in total) to represent each patient. The following procedure was then employed for identifying biallelic loss in each of the 781 cancer/HR associated genes. First, high-frequency germline SNV/indels (Supplementary Fig. 18) were marked as benign (*P*-score = 0). Then, each gene was screened for the following events: (i) a deep deletion; (ii) LOH in combination with a germline SNV/indel with a *P*-score ≥ 4 (likely pathogenic or higher in impact); (iii) LOH in combination with a somatic SNV/indel with a *P*-score ≥ 3 (VUS or higher in impact); or (iv) two SNVs/indels (germline + somatic, or 2x somatic) both with a *P*-score = 5 (pathogenic). See Supplementary Data 6 for details of the *P*-score thresholds used.

After applying CHORD to the HMF cohort, we then determined whether each of the 781 genes was significantly more frequently deficient in CHORD-HRD vs. CHORD-HRP patients using a one-tailed Fisher's exact test, with multiple testing correction performed with the p.adjust() function in R. This was done to determine the genes most likely to cause HRD when inactivated. Six genes were found with a *q*-value < 0.1 and had at least five patients with a deficiency in the corresponding gene: *BRCA1*, *BRCA2*, *RAD51C*, *PALB2*, *NF1*, and *STARD13* (Supplementary Fig. 19). *NF1* and *STARD13* have not been reported to be involved in HR, and thus further analyses were performed to validate the enrichment for these two genes.

Since *BRCA1* and *NF1* are both located on Chr17, we reasoned that copy number alterations (CNA; in this case referring to deep deletions or LOH) that affect *BRCA1* also affect *NF1*. This leads to frequent biallelic loss of *NF1* even though the gene is likely not associated with HRD. A similar situation was suspected for *BRCA2* and *STARD13* which are both located on Chr13. Thus, one-tailed Fisher's exact tests were performed to determine whether CNAs in each of the 781 genes significantly co-occurred more often with a CNA in *BRCA1* or *BRCA2*. Multiple testing correction was performed using the Hochberg procedure. Indeed, enrichment in the co-occurrence of *BRCA1* and *NF1* CNAs was found, and was similarly the case for *BRCA2* and *STARD13* (Supplementary Fig. 20). We thus concluded that biallelic loss of *NF1* and *STARD13* are likely not associated with HRD and were therefore excluded from Fig. 3a.

**Clustering of CHORD-HRD samples**. Clustering of CHORD-HRD samples based on biallelic inactivating events (as in Fig. 3c) is illustrated in Supplementary Fig. 25. First, samples were split into 4 groups according to their HRD subtype and whether a sample had an impactful biallelic event (*P*-score pair of 5 and ≥3).

For each of these groups, the HRD associated gene with the max *BP*-score was greedily determined per sample and assigned a score of 1, with the remaining genes being assigned a score of 0. Genes were prioritized as follows *BRCA2*, *BRCA1*, *RAD51C*, *PALB2*. This was based on highest to lowest enrichment of gene deficiency in CHORD-HRD vs. CHORD-HRP as described above. With the resultant (1,0) matrix, a sorting operation was performed. A post-processing step (done purely for cosmetic purposes) ensured that samples with deep deletions, LOH + frameshift, and LOH + other SNV/indels in the corresponding gene were ranked first. The sorted (1,0) matrices from the 4 sample groups were combined, and consecutive rows of 1s were considered a cluster. For the 2 groups representing samples with no impactful biallelic event, all samples were considered to be in one cluster. These 2 groups corresponded to clusters 4 and 6 in Fig. 3c, and samples in these clusters were considered to have an unknown cause of HRD.

For Supplementary Fig. 23, samples were first split by cancer type before performing the above procedure.

**Reporting summary**. Further information on research design is available in the Nature Research Reporting Summary linked to this article.

## Data availability

Metastatic WGS data and corresponding metadata have been obtained from the Hartwig Medical Foundation and provided under data request numbers DR-010 and DR-047. Both WGS data and metadata is freely available for academic use from the Hartwig Medical Foundation through standardized procedures and request forms can be found at https://www.hartwigmedicalfoundation.nl. WGS data for the 560 primary breast cancer (BRCA-EU) dataset and Pan-Cancer Analysis of Whole Genomes (PCAWG) primary cancer dataset are publically available from the International Cancer Genome Consortium (ICGC) (https://dcc.icgc.org/repositories; https://dcc.icgc.org/releases/PCAWG). For access to identifying data (e.g. germline or raw read data) for the PCAWG or BRCA-EU datasets, researchers will need to request access via the ICGC Data Access Compliance Office (DACO). All other data are available within the article, Supplementary Information or available from the authors upon request.

## Code availability

CHORD is available as an R package at https://github.com/UMCUGenetics/CHORD (DOI: 10.5281/zenodo.4020925). The code used for data processing and generating the figures is also available in this repository.

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

## Acknowledgements

This publication and the underlying study have been made possible partly on the basis of the data that Hartwig Medical Foundation and the Center of Personalised Cancer Treatment (CPCT, The Netherlands) have made available to the study. We thank Neeltje Steeghs (Netherlands Cancer Institute), Martijn Lolkema (Erasmus Medical Center Rotterdam), Geert Cirkel (Meander Medical Center), Els Witteveen (UMC Utrecht), Mariette Labots (Amsterdam UMC, location VUmc) and Laurens Beerepoot (Elisabeth-TweeSteden Ziekenhuis, Tilburg) for study inclusion of a significant part of the patients that were used in this study and Peter Bouwman (Netherlands Cancer Institute, Amsterdam) for critically reading the manuscript. This work was financially supported by the gravitation program CancerGenomiCs.nl from the Netherlands Organisation for Scientific Research (NWO) and Oncode Institute to E.C.

## Author contributions

L.N. performed analyses, wrote/edited the paper. J.M. edited the paper and provided discussion. A.V.H. conceived the study, performed analyses, wrote/edited the paper. E.C. edited the paper and provided discussion. E.C. and A.V.H. supervised the study. All authors proofread, made comments, and approved the paper.

## Competing interests

The authors declare no competing interests.
