## [Peer Review File · Nature Communications]

REVIEWER COMMENTS

Reviewer #1 (Remarks to the Author):

The authors have addressed my major technical concern relative to the confounding effects of treatment induced mutations. I have no further concerns.

Reviewer #2 (Remarks to the Author):

The points I raised have been acknowledged and either addressed or the sentence/point has been removed from the manuscript.

In summary, CHORD is a tool that uses near-identical principles to current state-of-the-art (HRDetect).

It does not however, offer technical improvement over previous state-of-the-art.

It does not offer new biological value.

Its application on metastatic cancers does not enhance biological knowledge and if anything serves to be a little misleading because in this cohort of cancers, it has not been possible to subtract mutations of the primary tumour.

At this point, it is another algorithm which is simply competing with currently available algorithms.

Is CHORD itself publishable as a method? Yes, somewhere.

Is it worthy of Nature Communications? In the absence of technical enhancement or biological insights, it is difficult to be convinced that it merits publication at Nature Comms. I still find it to be more of a specialist bioinformatics piece.

Reviewer #3 (Remarks to the Author):

In the revised manuscript, Nguyen et al have made several improvements.

Most notably, the CHORD algorithm itself was somewhat altered, such that it considers separately the deletions with microhomologies (MH) of different lengths. These MH length bins are informative for discriminating radiation-treated tumors, thus the more detailed feature set is a welcome addition. Because of these extra features, fewer tumors are declared HR-deficient (based on subclonal mutations).

This is ostensibly because the indels due to radiation treatment do not confound the classifier anymore: note that this was not shown with data, while it could have been. In particular, in Supplementary Fig 14b, are the false-positives that disappear in Supp Fig 14a those tumors that

were irradiated? Without showing this with data, it is hard to claim that "having the microhomology deletions feature split by these two homology length bins may mitigate radiotherapy-associated false positive predictions."

Next, we now have an application of the CHORD to an additional PCAWG dataset, which gives a more complete representation of the landscape of HR deficiencies of the BRCA1-type or BRCA2-type across cancer types, providing some insights (which are valuable even if preliminary) of the differences between primary and metastatic tumors. This is certainly a welcome addition to the manuscript.

Finally, there were multiple edits to the text to add references, and rephrase or delete speculative/less supported statements; again an improvement. Some suggestions/queries made by the reviewers were ignored, based on that they are out-of-scope. I think that's fine -- overall I would tend to agree with the authors in which points they declined to address, with one exception.

Major concern:

There is one important point which was not addressed adequately in this reviewer's opinion, and is of a technical rather than conceptual nature. Thus it would need attention before I can recommend publication. I am referring to the point 2a of Reviewer #3, which was largely echoed in point 2 by Reviewer #4.

This is an important concern about portability of their predictive model across cancer types. In brief, because the model was trained mostly on ovarian, breast and pancreatic HR-deficient cancers, it is not clear how confident one can be in the predictions/prevalence of HR deficiencies across other cancer types

Their response to this query was "We assume that regardless of the mechanism of inactivation of an HRD gene" [which could differ between tissues] "the mutational pattern will look similar.". It is precisely this assumption that was not tested rigorously yet will affect application of CHORD on many cancer types where HRD is not commonly appreciated.

They offered a tSNE plot in response to this point in the rebuttal document (it apparently was not included in the revised manuscript - please add the plot, or a revised version thereof). This tSNE plot was created based on HR-deficiency-predictive features, and so it (unsurprisingly) separates the HR-deficient tumors from the others.

However I think this tSNE analysis does not actually support their argument very strongly, because within the HR-deficient cluster, one can appreciate there are 4 sub-clusters. These 4 sub-clusters are enriched with different tissues. This suggests that HR-deficiency genomic features do differ somewhat across tissues, and thus there may be reason to worry about portability of their CHORD classifier across tissues.

If the authors wish to keep the pan-cancer CHORD (and not introduce multiple, tissue-specific CHORDs), I think we would at the very least need to see an estimate of error introduced when

generalizing across tissues, and (if possible) which tissues are more affected by such errors. These error estimates for a cancer type need to be presented as a possible caveat when reporting prevalence of HRD across cancer types.

This could shed light on the discrepancies such as e.g. the result Reviewer #4 commented on “Bone/Soft tissue sarcomas authors identified 2% HRD compared to previous estimates of 17% in Seligson et al, JCO, 2018, and in 27% in Gulhan et al, Nat Gen, 2019.”. One possible way to obtain such error estimates per-tissue is to perform cross-validation by leaving one tissue out, and then testing on that tissue (I appreciate that rarely-HR-deficient tissues may need to be pooled together for this to work).

Reviewer #4 (Remarks to the Author):

The authors have done a substantial amount of additional work for their revision. I don't think the queries that I have made were fully resolved, however.

The two main technical concerns were the following.

1. We raised the point that the SBS signature assignment process used by the authors is problematic. The authors argue in their response that "We do not claim that avoiding mutational signatures leads to improved HRD prediction, nor that it is better to use the 6 SNV types for HRD prediction. We do acknowledge that using more sophisticated methods for signature fitting (e.g. used in SigProfiler) could have resulted in better assignment of signature contributions."

However, as far as I can tell, the manuscript still makes the same claim (p.5): “SBS signature 3 (proposed as a sensitive marker for HRD in recent studies[15–17]) is actually a less important feature for predicting HRD than microhomology deletions in both HRDetect (as demonstrated by the authors [9]) and CHORD-signature, indicating that microhomology deletions serves as a better (univariate) marker of HRD compared to SBS signature 3 [10]. We thus conclude that accurate detection of HRD does not require mutational signatures, thereby simplifying HRD calling and avoiding the complications associated with the fitting step required for computing signature contributions in individual samples (for which there is currently no consensus approach)[18]”

Because the SBS assignment process is not updated after our inquiry and further calculations are not provided to support the above statement, such statements seem unwarranted.

2. We noted, as other reviewers also did, that the application of a classifier trained on a small set of tumor types to the pan-cancer setting may be problematic. The authors point to the tSNE calculation where the same input features were used and all the tumor types were pooled together and show that the HRD tumors based on their classifier cluster together. But I'm not sure if this calculation provides additional support. In fact, the tSNE figure, for examples, shows that 60% (3 out of 5) bone/soft tissue CHORD-HRD samples do not fall in the main HRD cluster. Authors should state in

the main text that the classification in tumor types that are underrepresented in the training process may have a different rate of accumulation of variants related to HRD and may not be classified accurately using this method.

Reviewer #1 (Remarks to the Author):

The authors have addressed my major technical concern relative to the confounding effects of treatment induced mutations. I have no further concerns.

Reviewer #2 (Remarks to the Author):

The points I raised have been acknowledged and either addressed or the sentence/point has been removed from the manuscript.

In summary, CHORD is a tool that uses near-identical principles to current state-of-the-art (HRDetect). It does not however, offer technical improvement over previous state-of-the-art.

We respectfully disagree with the reviewer. Although CHORD uses similar principles as other tools, its feature choices, training algorithm used and implementation involve many differences. Furthermore, CHORD does not rely on COSMIC signatures, which allows one to skip the non-trivial signature fitting step for computing signature contributions in individual samples (for which there is currently no consensus approach), which reduces feature noise leading to more accurate HRD predictions. Furthermore, CHORD has been trained on pan-cancer data and thus overcomes the small, but noticeable (Degasperi et al 2020), intertissue variation of mutation scars. This likely explains why CHORD did not miss a few HRD prostate samples from the PCAWG cohort compared to HRDetect (which was mainly trained on breast cancer data) and ultimately, does not hinder implementation in standard clinical workflows. Lastly, while the HRDetect code is currently available on github, it is not trivial to get up and running as it is built as a pipeline that is restricted to Linux-based platforms and requires command line knowledge to set up and use. On the other hand, CHORD can easily be installed from github directly within R (allowing it to be platform independent), after which HRD can be predicted from standard VCFs or tabular data as input using only 2 R functions (one to generate the mutational features and one for prediction). We believe that these combined features do justify the existence and publication of CHORD alongside other tools in this area.

It does not offer new biological value.

Again, we respectfully disagree with the reviewer. We believe the systematic pan-cancer overview, the demonstration that the tool works on metastatic cancer samples, and the analysis of underlying genetic defects all provide additional value over existing data. Even when considering the scientific advances limited (for which we disagree), the validation of previous findings with independent tools and in independent datasets does in our view also justify scientific publication.

Its application on metastatic cancers does not enhance biological knowledge and if anything serves to be a little misleading because in this cohort of cancers, it has not been possible to subtract mutations of the primary tumour.

We do not aim to draw conclusions on tumor evolution from primary to metastatic as it is indeed not possible to reliably distinguish mutations that occurred in the primary or metastatic phase of the cancer.

Our results only describe the HRD landscape in representative primary and metastatic pan-cancer cohorts. The relevance for doing this with metastatic cancer data as input is in our view very high as comprehensive molecular tumor characterizations in the clinic is often done at the metastatic stage of the disease.

At this point, it is another algorithm which is simply competing with currently available algorithms.

Is CHORD itself publishable as a method? Yes, somewhere.

Is it worthy of Nature Communications? In the absence of technical enhancement or biological insights, it is difficult to be convinced that it merits publication at Nature Comms. I still find it to be more of a specialist bioinformatics piece.

Reviewer #3 (Remarks to the Author):

In the revised manuscript, Nguyen et al have made several improvements.

Most notably, the CHORD algorithm itself was somewhat altered, such that it considers separately the deletions with microhomologies (MH) of different lengths. These MH length bins are informative for discriminating radiation-treated tumors, thus the more detailed feature set is a welcome addition. Because of these extra features, fewer tumors are declared HR-deficient (based on subclonal mutations).

This is ostensibly because the indels due to radiation treatment do not confound the classifier anymore: note that this was not shown with data, while it could have been. In particular, in Supplementary Fig 14b, are the false-positives that disappear in Supp Fig 14a those tumors that were irradiated? Without showing this with data, it is hard to claim that "having the microhomology deletions feature split by these two homology length bins may mitigate radiotherapy-associated false positive predictions."

We compared the predictions from CHORD vs CHORD-del.mh.merged (the version of the algorithm where we do not make a distinction between deletions with different MH lengths) on all variants detected in a sample (see below figure) and found that all 5 samples predicted HRD only by CHORD-del.mh.merged had radiotherapy pretreatment, while of the 9 samples predicted HRD only by CHORD, 5 had radiotherapy though 2 had evidence of BRCA1/2 biallelic loss.

These data suggest that splitting microhomology deletions into two homology length bins may reduce false positive predictions due to radiotherapy slightly, although the low number of discrepant samples between CHORD and CHORD-del.mh.merged also indicates that the impact of radiotherapy on HRD prediction is minimal when using all somatic variants (clonal plus subclonal) as input (which is likely the default setting for routine application). We have now included the below figure in the manuscript as **Supplementary figure 16**, and adjusted the text under 'Effect of treatment on HRD predictions' to better reflect the impact of splitting microhomology deletions into two bins.

Next, we now have an application of the CHORD to an additional PCAWG dataset, which gives a more complete representation of the landscape of HR deficiencies of the BRCA1-type or BRCA2-type across cancer types, providing some insights (which are valuable even if preliminary) of the differences between primary and metastatic tumors. This is certainly a welcome addition to the manuscript.

Finally, there were multiple edits to the text to add references, and rephrase or delete speculative/less supported statements; again an improvement. Some suggestions/queries made by the reviewers were ignored, based on that they are out-of-scope. I think that's fine -- overall I would tend to agree with the authors in which points they declined to address, with one exception.

Major concern:

There is one important point which was not addressed adequately in this reviewer's opinion, and is of a technical rather than conceptual nature. Thus it would need attention before I can recommend publication. I am referring to the point 2a of Reviewer #3, which was largely echoed in point 2 by Reviewer #4.

This is an important concern about portability of their predictive model across cancer types. In brief, because the model was trained mostly on ovarian, breast and pancreatic HR-deficient cancers, it is not

clear how confident one can be in the predictions/prevalence of HR deficiencies across other cancer types

Their response to this query was "We assume that regardless of the mechanism of inactivation of an HRD gene" [which could differ between tissues] ", the mutational pattern will look similar.". It is precisely this assumption that was not tested rigorously yet will affect application of CHORD on many cancer types where HRD is not commonly appreciated.

They offered a tSNE plot in response to this point in the rebuttal document (it apparently was not included in the revised manuscript - please add the plot, or a revised version thereof). This tSNE plot was created based on HR-deficiency-predictive features, and so it (unsurprisingly) separates the HR-deficient tumors from the others.

However I think this tSNE analysis does not actually support their argument very strongly, because within the HR-deficient cluster, one can appreciate there are 4 sub-clusters. These 4 sub-clusters are enriched with different tissues. This suggests that HR-deficiency genomic features do differ somewhat across tissues, and thus there may be reason to worry about portability of their CHORD classifier across tissues.

If the authors wish to keep the pan-cancer CHORD (and not introduce multiple, tissue-specific CHORDs), I think we would at the very least need to see an estimate of error introduced when generalizing across tissues, and (if possible) which tissues are more affected by such errors. These error estimates for a cancer type need to be presented as a possible caveat when reporting prevalence of HRD across cancer types.

This could shed light on the discrepancies such as e.g. the result Reviewer #4 commented on "Bone/Soft tissue sarcomas authors identified 2% HRD compared to previous estimates of 17% in Seligson et al, JCO, 2018, and in 27% in Gulhan et al, Nat Gen, 2019.". One possible way to obtain such error estimates per-tissue is to perform cross-validation by leaving one tissue out, and then testing on that tissue (I appreciate that rarely-HR-deficient tissues may need to be pooled together for this to work).

As suggested by reviewer #3, we have trained CHORD-like models where cancer types were held out from the training set (which consisted of HMF samples). These models were then applied to the held out HMF samples as well as the PCAWG samples (as independent data) to calculate the likely prediction error for each cancer type (see below). Using the default classification cutoff of 0.5, we observed overall a low false positive rate (<2%) and false negative rate (<6%). The false negative rate was higher in biliary, lung and other cancer types, though these error estimates may not be entirely accurate due to the very low number of BRCA1/2 deficient samples in these cancer types. Notably, bone/soft tissue cancer had no misclassification suggesting that we do not underestimate the frequency of HRD in this cancer type (although other confounding factors may contribute to the discrepancy of our results with those of Seligson et al and Gulhan et al). These results thus indicate that CHORD likely has no or only minor cancer type bias and supports the notion that the mutational footprint of HRD is likely not cancer type

specific (as already suggested by the t-SNE clustering). We have included the t-SNE plot and the below figure in the revised manuscript as **Supplementary figure 8** and **9**, respectively.

Reviewer #4 (Remarks to the Author):

The authors have done a substantial amount of additional work for their revision. I don't think the queries that I have made were fully resolved, however.

The two main technical concerns were the following.

1. We raised the point that the SBS signature assignment process used by the authors is problematic. The authors argue in their response that "We do not claim that avoiding mutational signatures leads to improved HRD prediction, nor that it is better to use the 6 SNV types for HRD prediction. We do acknowledge that using more sophisticated methods for signature fitting (e.g. used in SigProfiler) could have resulted in better assignment of signature contributions."

However, as far as I can tell, the manuscript still makes the same claim (p.5): "SBS signature 3 (proposed as a sensitive marker for HRD in recent studies[15–17]) is actually a less important feature for predicting HRD than microhomology deletions in both HRDetect (as demonstrated by the authors [9]) and CHORD-signature, indicating that microhomology deletions serves as a better (univariate) marker of HRD

compared to SBS signature 3 [10]. We thus conclude that accurate detection of HRD does not require mutational signatures, thereby simplifying HRD calling and avoiding the complications associated with the fitting step required for computing signature contributions in individual samples (for which there is currently no consensus approach)[18]”

Because the SBS assignment process is not updated after our inquiry and further calculations are not provided to support the above statement, such statements seem unwarranted.

We agree and have removed the sentence “SBS signature 3 (proposed as a sensitive marker for HRD...” from the manuscript.

2. We noted, as other reviewers also did, that the application of a classifier trained on a small set of tumor types to the pan-cancer setting may be problematic. The authors point to the tSNE calculation where the same input features were used and all the tumor types were pooled together and show that the HRD tumors based on their classifier cluster together. But I'm not sure if this calculation provides additional support. In fact, the tSNE figure, for examples, shows that 60% (3 out of 5) bone/soft tissue CHORD-HRD samples do not fall in the main HRD cluster. Authors should state in the main text that the classification in tumor types that are underrepresented in the training process may have a different rate of accumulation of variants related to HRD and may not be classified accurately using this method.

See response to major concern of reviewer #3.

REVIEWERS' COMMENTS

Reviewer #3 (Remarks to the Author):

The authors have addressed my queries regarding (i) portability of CHORD across cancer types and regarding (ii) the role of MH length features satisfactorily. Therefore I can recommend publication of the manuscript in this journal: the method and the analyses are sound and they represent an advance over the state-of-the-art; additionally the CHORD tool is likely to be useful for downstream research work and also potentially in translational applications.

Reviewer #4 (Remarks to the Author):

The authors are failing to understand the reviewer's point. The held-out analysis is not sufficient to support a pan-cancer application of the CHORD classifier. The analysis showed that the training set is somewhat homogeneous across tissue types, which is not unexpected, since the true positive HRD samples are defined as those with BRCA mutations. However, our concern is exactly about the training set itself -- HRD samples that are not caused by BRCA loss may be under-represented. As a result, the CHORD classifier may be forced to identify BRCA-related samples, instead of more generic HRD ones. This problem is particularly relevant for tumor types that do not generally have frequent BRCA mutations, and may underlie the discrepancies in HRD frequencies as mentioned in previous rounds of reviews.

At this point, there has been enough back and forth and I am reluctantly supportive of the paper's publication. However, I insist that the authors acknowledge this caveat in the manuscript, given the clinical implications the result may have.